# Association of Oral Anticoagulation Prescription with Clinical Events in Patients with an Asymptomatic Unrepaired Abdominal Aortic Aneurysm

**DOI:** 10.3390/biomedicines10092112

**Published:** 2022-08-29

**Authors:** Jonathan Golledge, Jason Jenkins, Michael Bourke, Bernard Bourke, Tejas P. Singh

**Affiliations:** 1Queensland Research Centre for Peripheral Vascular Disease, College of Medicine and Dentistry, James Cook University, Townsville, QLD 4811, Australia; 2The Department of Vascular and Endovascular Surgery, Townsville University Hospital, Townsville, QLD 4812, Australia; 3The Australian Institute of Tropical Health and Medicine, James Cook University, Townsville, QLD 4811, Australia; 4Royal Brisbane and Women’s Hospital, Herston, QLD 4006, Australia; 5Gosford Vascular Clinic, Gosford, NSW 2250, Australia

**Keywords:** abdominal aortic aneurysm, surgical repair, rupture and anticoagulation

## Abstract

Background: Most abdominal aortic aneurysms (AAA) have large volumes of intraluminal thrombus which has been implicated in promoting the risk of major adverse events. The aim of this study was to examine the association of therapeutic anticoagulation with AAA-related events and major adverse cardiovascular events (MACE) in patients with an unrepaired AAA. Methods: Patients with an asymptomatic unrepaired AAA were recruited from four sites in Australia. The primary outcome was the combined incidence of AAA repair or AAA rupture-related mortality (AAA-related events). The main secondary outcome was MACE (the combined incidence of myocardial infarction, stroke, or cardiovascular death). The associations of anticoagulation with these outcomes were assessed using Cox proportional hazard analyses (reporting hazard ratio, HR, and 95% confidence intervals, CI) to adjust for other risk factors. Results: A total of 1161 patients were followed for a mean (standard deviation) of 4.9 (4.0) years. Of them, 536 (46.2%) patients had a least one AAA-related event and 319 (27.5%) at least one MACE. In the sample, 98 (8.4%) patients were receiving long-term therapeutic anticoagulation using warfarin (84), apixaban (7), rivaroxaban (6), or dabigatran (1). Prescription of an anticoagulant was associated with a reduced risk of an AAA-related event (adjusted HR 0.61; 95% CI 0.42, 0.90, *p* = 0.013), but not MACE (HR 1.16; 95% CI 0.78, 1.72, *p* = 0.476). Conclusions: These findings suggest that AAA-related events but not MACE may be reduced in patients prescribed an anticoagulant medication. Due to the inherent biases of observational studies, a randomized controlled trial is needed to assess whether anticoagulation reduces the risk of AAA-related events.

## 1. Introduction

Worldwide abdominal aortic aneurysm (AAA) is estimated to be responsible for approximately 200,000 deaths annually due to aneurysm rupture or complications of surgical repair [1]. Patients with AAA are also at high risk of other major adverse cardiovascular events (MACE), including myocardial infarction and stroke [2]. Currently, AAA management focuses on surgical treatment to limit aneurysm rupture but there is growing interest in discovering effective medical management [3].

Most AAAs have large volumes of intraluminal thrombus [4,5]. AAA thrombus has been shown to contain high concentrations of proteolytic enzymes and pro-inflammatory cytokines implicated in AAA pathogenesis [3,6]. Large AAA thrombus volume and high concentrations of biomarkers of AAA thrombus, such as D-dimer, have been associated with fast aneurysm growth and an increased risk of MACE [4,7,8,9,10]. Some experimental studies have suggested that inhibiting thrombus formation can limit AAA growth and rupture [11,12,13]. There has, however, been limited clinical investigation of whether anticoagulation is beneficial for reducing the risk of AAA-related major adverse events (i.e., requirement for aneurysm repair or rupture) or MACE [12]. As a result the prescription of anticoagulants is not currently recommended in patients with AAA unless they have another indication such as atrial fibrillation or venous thrombosis [14,15].

In the absence of prior evidence, the aim of this observational study was to examine whether anticoagulant drug prescription, for concurrent indications, was associated with a reduced risk of AAA-related events and MACE in patients with asymptomatic unrepaired AAAs. It was aimed to obtain further insight as to whether a randomized control trial of anticoagulation might be worthwhile in AAA patients.

## 2. Methods

### 2.1. Study Design and Participants

This study was part of a prospective cohort study which commenced in 2002 aimed at identifying predictors of outcomes for patients with vascular disease [16,17]. The current study included patients with an asymptomatic unrepaired AAA with a diameter ≥ 30 mm who were recruited between February 2002 and November 2020 from four sites in Australia, including The Royal Brisbane and Women’s Hospital, Gosford Vascular Clinic, The Townsville University Hospital, and the Mater Hospital Townsville [10,11,12]. At the time of recruitment, participants were interviewed, medical records and imaging were reviewed, and risk factors and medications recorded on standard case report forms prior to entry into a database. Data collected included age, sex, diabetes, hypertension, ischemic heart disease, prior stroke, smoking history and medication prescriptions. Patients receiving warfarin or a non-vitamin K antagonist oral anticoagulant, including rivaroxaban, apixaban, and dabigatran, for long-term anticoagulation were identified and the reason for treatment recorded. For this study, any participants prescribed anticoagulation where the intention was to stop this in the future or who did not attend any follow-up visits were excluded. Diabetes, hypertension, ischemic heart disease, and prior stroke were defined based on prior diagnosis or treatment as documented in medical records. Infra-renal aortic diameter was measured from ultrasound or computed tomography imaging performed at the time of recruitment using reproducible protocols previously reported [16]. The study was approved by Townsville Hospital and Health Services Ethics committee (HREC/14/QTHS/203) and governance authorities and written informed consent was obtained from participants. 

### 2.2. Follow-Up and Outcome Assessment

Participants were followed up as part of their standard care according to protocols at each site. In general, this involved review annually if aortic diameter was 30 to 44 mm and every three to six months if aortic diameter was ≥45 mm [17]. The primary outcome for this study was the combined incidence of AAA repair (open or endovascular) or mortality due to AAA rupture, defined as AAA-related events [17]. The secondary outcomes were the combined incidence of non-fatal myocardial infarction, non-fatal stroke and cardiovascular death, myocardial infarction alone, stroke alone, and all-cause mortality [2]. For patients not experiencing an outcome event, follow-up was censored at the date of last review. Decisions about requirement for AAA repair were at the discretion of the treating consultant vascular surgeons, not standardized for the purpose of this study, and were independent of any planned analysis. 

### 2.3. Statistical Analyses

The risk factors and medications recorded at the time of recruitment were compared between groups using two tailed independent or Chi-squared tests. The freedom from primary and secondary outcome events were compared between different groups using Kaplan–Meier analysis with log rank test and Cox proportional hazard analyses using the SPSS v.25 software package (IBM) and Stata v16.1 (StataCorp LP, College Station, TX, USA). In the Cox proportional hazard analyses, a number of different models were created. These included unadjusted models and models adjusted for age and sex alone, established risk factors for AAA-related events (sex, current smoking, diabetes, hypertension, and initial aortic diameter) or MACE (age, sex, current smoking, diabetes, hypertension, ischemic heart disease, stroke, and initial aortic diameter), and risk factors identified to be disparate between participants who were and were not prescribed anticoagulant medication (based on *p* < 0.05 in univariate analysis). All models presented conformed to the proportional hazards assumption. For all analyses, *p*-values < 0.05 was considered statistically significant. 

### 2.4. Data Sharing

Original data are available from the corresponding author.

## 3. Results

### 3.1. Characteristics of Participants at Recruitment in Relation to Anticoagulant Drug Prescription

A total of 1161 patients were included, of whom 98 (8.4%) were prescribed a drug for therapeutic anticoagulation, including warfarin (n = 84), apixaban (n = 7), rivaroxaban (n = 6), or dabigatran (n = 1). The reasons for anticoagulation included atrial fibrillation or other arrhythmia (n = 69), previous valve replacement (n = 12), recurrent venous thrombosis (n = 13), or multiple reasons (n = 4), including atrial fibrillation and recurrent venous thrombosis (n = 2), valve replacement, atrial fibrillation and previous venous thrombosis (n = 1), and aortic valve replacement and atrial fibrillation (n = 1).

Participants prescribed an anticoagulant medication were more likely to have a diagnosis of ischemic heart disease and be prescribed a statin and frusemide for treating heart failure, but less likely to be receiving aspirin (Table 1). Other risk factors assessed were not significantly different between participants who were and were not prescribed an anticoagulant medication (Table 1).

### 3.2. Outcome Events

During a mean (standard deviation) follow-up of 4.9 (4.0) years, 536 (46.2%) participants had at least one AAA-related event including endovascular repair in 301 (25.9%), open repair in 222 (19.1%), and death due to AAA rupture in 14 (1.2%). In addition, 319 (27.5%) participants had a MACE, 119 (10.2%) had a myocardial infarction, 65 (5.6%) a stroke, and 436 (37.6%) died (224 due to cardiovascular causes). 

### 3.3. Association of Outcome Events with Anticoagulant Drug Prescription

AAA-related events were less common in participants prescribed an anticoagulant drug (32 of 98; 32.7%) compared to those not prescribed anticoagulants (504 of 1063; 47.4%; *p* = 0.005; Table 1). By Kaplan–Meier analysis, the freedom from AAA events were 81.2%, 67.5%, and 62.4% after 1, 3, and 5 years in participants prescribed an anticoagulant medication compared to 65.6%, 55.3%, and 48.6% after 1, 3, and 5 years in participants not prescribed an anticoagulant medication (*p* = 0.006 by log rank test; Figure 1). The frequency of MACE, myocardial infarction alone, stroke alone, cardiovascular death alone, and all-cause mortality were similar in the two participant groups (Table 1). Cox proportional hazard analyses showed that the risk of AAA-related events, but not MACE, was significantly lower in participants prescribed an anticoagulant drug compared to those not receiving this medication in both unadjusted and adjusted models (Table 2 and Table 3). These analyses included adjustment for potential confounding variables such as age, sex, initial AAA diameter, smoking, diabetes, hypertension, ischemic heart disease, prior stroke, and other prescribed medications (aspirin, statins, frusemide, and other antiplatelet medications). 

Graphs show cumulative proportions over 5 years. The freedom from AAA events was significantly greater in patients prescribed anticoagulants than those who were not (log-rank test *p* = 0.006). The freedom from MACE was not significantly different between groups (log-rank test *p* = 0.251). 

## 4. Discussion

The main finding from this study was that amongst patients with unrepaired AAAs, those who were receiving therapeutic anticoagulation had a lower risk of AAA-related events. This association was independent of established risk factors for AAA progression such as initial AAA diameter and smoking, and also potential confounding factors associated with anticoagulant prescription such as the prescription of statins and ischemic heart disease [4,13]. Due to the observational nature of this investigation, the risks associated with therapeutic anticoagulation and the relatively small number of participants receiving anticoagulation in this study, this finding should be interpreted cautiously prior to replication in other populations and a randomized controlled trial. 

Thrombus burden, as measured by circulating D-dimer levels, has been associated with the risk of cardiovascular events, such as myocardial infarction, in a number of populations such as patients with acute coronary syndrome [18]. A previous study of 98 patients with unrepaired AAA examined by computed tomography reported that participants with AAA thrombus volume above median had an increased risk of cardiovascular events (relative risk 2.8, 95% CI 1.01, 5.24) [4]. The current study is the first observational study to examine the association of therapeutic anticoagulation with MACE in patients with unrepaired AAA that we identified. The risk of MACE was not reduced in participants receiving therapeutic anticoagulation as compared with those who were not prescribed these drugs. The finding does not support the use of anticoagulation to reduce the risk of MACE in patients with unrepaired AAA outside other specific indications, such as stroke prevention in patients with arrhythmia.

A large body of preclinical research has implicated AAA thrombus as responsible for driving AAA progression through a variety of mechanisms including promoting aortic wall inflammation, releasing proteolytic enzymes, and depriving the aortic tunica media of oxygen [3,6,12]. Studies in the elastase, angiotensin II, and xenograft animal models of AAA have suggested that blocking aneurysm thrombus formation reduces aneurysm progression [6,11,12,13]. Furthermore, large size of AAA thrombus on imaging and higher circulating concentrations of D-dimer (due to thrombus remodeling) have been associated with more rapid AAA growth in patients [4,7,8,9,10]. Incubation of human AAA explants with rivaroxaban in vitro has been reported to improve aortic wall mitochondrial function, reduce expression of markers of mitophagy, reduce expression of matrix metalloproteinase-9, and limit release of interleukin-6, which have all been implicated in AAA growth [19,20]. Together, these data provide a strong rationale for expecting that reducing AAA thrombosis might reduce clinically important events in patients with unrepaired AAA.

A number of observational studies and one randomized controlled trial has examined the role of antiplatelet medications in patients with unrepaired AAA [21,22,23]. One large (n = 8020) cross-sectional registry study in Demark reported that aspirin prescription was associated with a lower risk of presenting with AAA rupture (odds ratio 0.72, 95% CI 0.66, 0.79) but after adjustment for potential confounding risk factors, there was no reduction in risk of rupture (odds ratio 0.97, 95% CI 0.86, 1.08) [21]. Another Danish study reported that the growth of AAAs measuring 40 to 49 mm (but not <40 mm) was significantly slower in patients prescribed low dose aspirin than those not receiving this medication [22]. A randomized controlled trial testing the anti-platelet medication ticagrelor in 144 patients with 35 to 49 mm AAAs reported no significant effect on increase in AAA diameter or volume over 12 months [23]. It should be noted that ticagrelor also had no significant effect on AAA thrombus volume. Overall, these findings suggest that anti-platelet drugs are not effective at limiting AAA progression excepting the absence of a large randomized controlled trial.

Previous observational studies have associated anticoagulant drug prescription with faster sac expansion after endovascular aneurysm repair [24,25]. This has been suggested to be due to higher rates of endoleak and persistent AAA sac perfusion in patients receiving anticoagulation due to reduced thrombosis [24,25,26]. The mechanisms responsible for progression of unrepaired AAAs are thought to be completely different to sac expansion following endovascular repair though. There has been limited investigation of the effect or association of anticoagulants, including warfarin, rivaroxaban, or other non-vitamin K antagonist oral anticoagulant, with AAA growth or AAA-related events in patients with unrepaired AAAs. A recently published nationwide analysis of hospital data including 3.8 million patients suggested that AAA patients prescribed an anticoagulant were less likely to experience AAA rupture during follow-up (Odds ratio: 0.92; 95% CI: 0.87 to 0.98; *p* = 0.01) [27]. That study had several limitations, however, including its retrospective design and use of hospital coding data. The current study therefore appears to represent the first prospective observational investigation of the association of anticoagulation with AAA-related events in this population. The finding of an approximate 40% relative reduction in AAA-related events is promising but a number of limitations of the study should be acknowledged. Firstly, while this study included one of the largest group of patients with unrepaired AAA in which AAA-related events have been studied, only 98 were prescribed anticoagulants. Participants were recruited from four Australian sites and thus may not reflect patients attending other centers in Australia or those overseas. Of note, Australia does not have a screening program for AAA. The included aneurysms were larger than reported in many screening programs. Thus, the findings may not be generalizable to other populations. Drug history and compliance were not routinely collected during follow-up. While the efficacy of anticoagulation by warfarin (using the international normalized ratio) was monitored in relevant patients by their general practitioner or local laboratories, this was not recorded as part of the study and thus not available to analyze. The decision to perform AAA repair was at the discretion of the treating consultant vascular surgeon and was not standardized. Furthermore, independent adjudication of events was not performed so it is possible that inaccuracies may have occurred. Observational studies are subject to confounding and selection bias. We attempted to reduce the influence of confounding by adjusting for risk factors for AAA-related events and differences between patient groups. It however remains impossible to completely exclude residual confounding. 

In conclusion, this study found that over approximately 5 years, about half of the patients with an unrepaired aortic aneurysm had an AAA-related event, while one-quarter had a MACE. The risk of an AAA event was lowered by approximately 40% in patients that were therapeutically anticoagulated. The risk of MACE was not affected. The findings suggest that anticoagulation should be examined as a potential AAA treatment in other populations and if these observational findings are replicated, a randomized controlled trial may be warranted to test anticoagulation as a treatment for AAA.

## Figures and Tables

**Figure 1 biomedicines-10-02112-f001:**
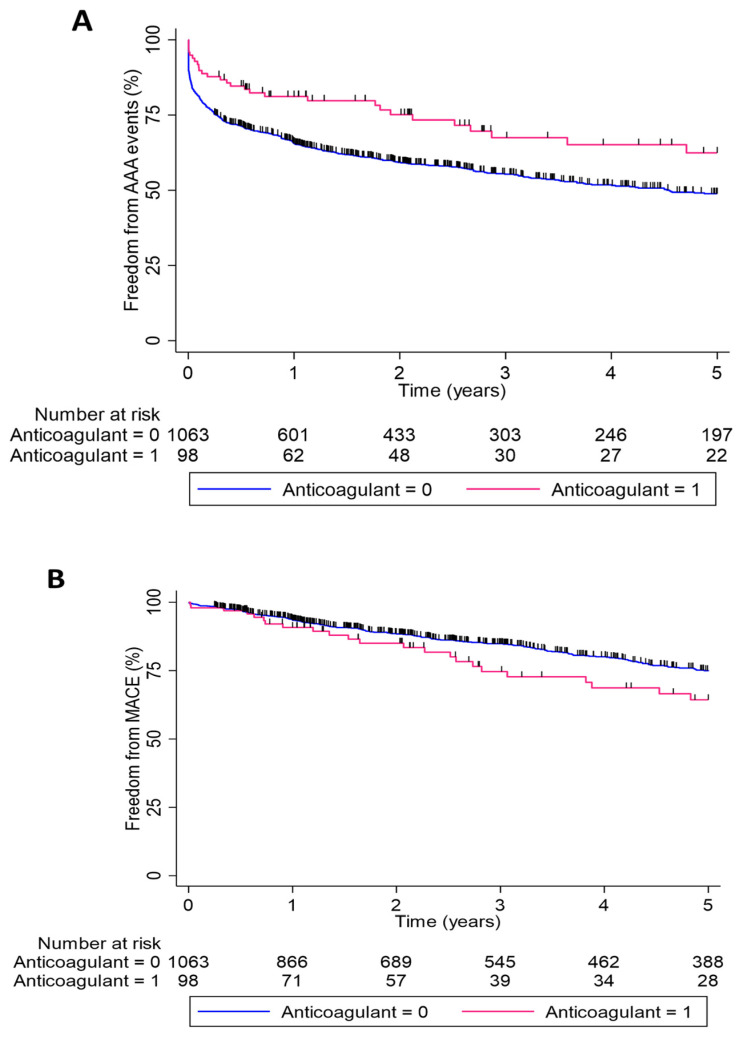
Freedom from AAA repair or mortality due to AAA rupture (AAA events; (**A**)) and major adverse cardiovascular events (MACE; (**B**)) in patients prescribed anticoagulation (pink line) and those not prescribed these mediations (blue line).

**Table 1 biomedicines-10-02112-t001:** Characteristics of patients in relation to prescription of an anticoagulant.

Factor	Prescribed Anticoagulant		*p* Value
	Yes	No	
Number	98	1063	
Age (years)	73.9 ± 7.6	73.0 ± 7.7	0.247
Female	17 (17.4%)	188 (17.7%)	0.933
Smoking			0.258
Current	22 (22.5%)	282 (26.5%)	
Former	67 (68.4%)	640 (60.2%)	
Never	9 (9.2%)	141 (13.3%)	
Diabetes	26 (26.5%)	219 (20.6%)	0.169
Hypertension	81 (82.7%)	801 (75.4%)	0.106
Ischemic heart disease	76 (77.6%)	504 (47.4%)	<0.001
Family history of aortic or peripheral aneurysm	11 (11.2%)	108 (10.2%)	0.740
Stroke	15 (15.3%)	101 (9.5%)	0.067
Aspirin	32 (32.7%)	649 (61.1%)	<0.001
Other antiplatelet	10 (10.2%)	180 (16.9%)	0.085
Statin	74 (75.5%)	693 (65.2%)	0.039
Metformin	12 (12.2%)	127 (12.0%)	0.931
Frusemide prescribed for cardiac failure	25 (25.5%)	79 (7.4%)	<0.001
Body mass index *	28.2 ± 4.9	27.6 ± 5.0	0.251
Initial AAA diameter	44.2 ± 10.3	46.5 ± 11.4	0.053
Follow-up (years)	4.5 ± 4.3	4.9 ± 4.0	0.299
AAA repair or AAA rupture related mortality	32 (32.7%)	504 (47.4%)	0.005
Major adverse cardiovascular events	30 (30.6%)	289 (27.2%)	0.467
Myocardial infarction	10 (10.2%)	109 (10.3%)	0.988
Stroke	6 (6.1%)	59 (5.6%)	0.814
Cardiovascular death	22 (22.4%)	202 (19.0%)	0.408
All-cause death	42 (42.9%)	394 (37.1%)	0.257

Shown are number (percentage) or mean (±standard deviation). Missing from 123* patients. *p* values are from Chi-squared or two tailed independent *t* tests.

**Table 2 biomedicines-10-02112-t002:** Association of anticoagulant prescription with AAA-related events.

Model	Hazard Ratio	95% Confidence Intervals	*p* Value
Unadjusted	0.61	0.43, 0.87	0.007
Adjusted *	0.62	0.43, 0.89	0.010
Adjusted †	0.62	0.43, 0.89	0.010
Adjusted ‡	0.61	0.42, 0.90	0.013

* Age and sex; † sex, current smoking, diabetes, hypertension, and initial aortic diameter; ‡ ischemic heart disease, prior stroke, aspirin, other antiplatelet medications, statins, frusemide, initial aortic diameter.

**Table 3 biomedicines-10-02112-t003:** Association of anticoagulant prescription with major adverse cardiovascular events.

Model	Hazard Ratio	95% Confidence Intervals	*p* Value
Unadjusted	1.25	0.86, 1.82	0.252
Adjusted *	1.27	0.87, 1.85	0.220
Adjusted †	1.09	0.74, 1.59	0.672
Adjusted ‡	1.16	0.78, 1.72	0.476

* Age and sex; † age, sex, current smoking, diabetes, hypertension, Ischemic heart disease, stroke, initial aortic diameter; ‡ ischemic heart disease, stroke, aspirin, other antiplatelet medications, statins, frusemide, initial aortic diameter.

## Data Availability

Requests for data should be addressed to the corresponding author.

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
