# Peer review of "Association of Oral Anticoagulation Prescription with Clinical Events in Patients with an Asymptomatic Unrepaired Abdominal Aortic Aneurysm"

_biomedicines, 2022, doi:10.3390/biomedicines10092112_

Round 1

Reviewer 1 Report

The paper titled: Association of oral anticoagulants with clinical events in patients with asymptomatic abdominal aortic aneurysms; concentrates on the impact of anticoagulant applied by the patients with AAA.

Although the paper is interesting, I have some major concerns:

Methods

Authors investigated patients from different centres: The Royal Brisbane and Women’s Hospital, Gosford Vascular Clinic, The Townsville University Hospital and the Mater Hospital Townsville. There is no information about the potential impact of this aspect on the presented results.

Conclusions

Conclusions should be extended and related to the results. Currently it is short (5 lines) paragraph that presents no highlights for the manuscript.

Figure 1 is unreadable. Font size should increase. Marks placed on the figures are not visible. The size should be increased. Perhaps Authors should consider application not perpendicular configuration but vertical configuration.

Author Response

  1. Although the paper is interesting, I have some major concerns:

We thank the reviewer for assessing our paper.

  1. Methods

Authors investigated patients from different centres: The Royal Brisbane and Women’s Hospital, Gosford Vascular Clinic, The Townsville University Hospital and the Mater Hospital Townsville. There is no information about the potential impact of this aspect on the presented results.

The following has been added to the discussion on page 8:

Firstly, while this study included one of the largest group of patients with unrepaired AAA in which AAA related events have been studied, only 98 were prescribed anticoagulants. Participants were recruited from four Australian sites and thus may not reflect patients attending other centers in Australia or those overseas. Of note, Australia does not have a screening program for AAA. The included aneurysms were larger than reported in many screening programs. Thus the findings may not be generalizable to other populations.

  1. Conclusions

Conclusions should be extended and related to the results. Currently it is short (5 lines) paragraph that presents no highlights for the manuscript.

This section has been extended as requested and now reads as follows (page 8):

In conclusion, this study found that over approximately 5 years about half of the patients with an unrepaired aortic aneurysm had an AAA related event, while one-quarter had a MACE. The risk of an AAA event was lowered by approximately 40% in patients that were therapeutically anticoagulated. The risk of MACE was not affected. The findings suggest that anticoagulation should be examined as a potential AAA treatment in other populations and if these observational findings are replicated a randomized controlled trial may be warranted to test anticoagulation as a treatment for AAA.

  1. Figure 1 is unreadable. Font size should increase. Marks placed on the figures are not visible. The size should be increased. Perhaps Authors should consider application not perpendicular configuration but vertical configuration.

The figure has been revised as suggested (please see the revised paper).

Reviewer 2 Report

In this multicenter observational study authors found that AAA related events are lower in patients with unrepaired AAAs that are therapeutically anticoagulated for medical reasons other than the existence itself of an AAA. This study has numerous limitations that are acknowledged by the authors themselves. These data can therefore be considered as hypothesis generators. Further studies on a larger scale and with randomization of enrolled cases are needed to validate or not validate their results. Having said that, the present study presents interesting data collected, analyzed and thoroughly exposed. 

Author Response

We thank the reviewer for assessing our paper and the acknowledgement that we have explained the limitations of the research. We have made a number of textual revisions throughout the paper aimed to improve the ease of reading the article.

Round 2

Reviewer 1 Report

The authors corrected the manuscript and it is suitable for publication.